# Tailored Therapeutic Strategies for Fetuses, Neonates, Pediatrics, Geriatrics, Athletes, and Critical Cases in the Era of Personalized Medicine

**DOI:** 10.3390/diseases14010012

**Published:** 2025-12-29

**Authors:** Ahmed Bakr, Youssef Basem, Abanoub Sherif, Alamer Ata, Nada Nabil Saad, Yassmin Emarh Fayed, Maria Tamer, Malak Nasr Elkady, Rehab Abdelmonem

**Affiliations:** 1Industrial Pharmacy Department, College of Pharmaceutical Sciences and Drug Manufacturing, Misr University for Science and Technology (MUST), 6th of October City, Giza 12566, Egypt; 2Medical and Pharmaceutical Industrial Biotechnology Department, College of Biotechnology, Misr University for Science and Technology (MUST), 6th of October City, Giza 12566, Egypt; 200025117@must.edu.eg (Y.B.);; 3Biotechnology Department, Faculty of Agriculture, Cairo University, Giza 12566, Egypt; 11612021100428@stud.cu.edu.eg (N.N.S.);; 4Department of Biotechnology, Faculty of Biotechnology, Misr University for Science and Technology, 6th of October City, Giza 12566, Egypt

**Keywords:** precision medicine, multi-omic, phenotypic, targeted oncology, paediatrics, training, injury-risk stratification

## Abstract

Precision medicine, which relies on genomic, multi-omic, phenotypic, and environmental data, has the potential to transform healthcare from population-focused heuristics to individualized prevention, diagnosis, and treatment. Moreover, recent advances in sequencing, molecular profiles, wearable sensors, and machine learning have created opportunities for rapid translational innovation: rapid genomic diagnosis in neonatal and paediatric rare diseases, targeted oncology, pharmacogenomic-based prescribing strategies, and individual sport performance. Nevertheless, the vast majority of innovations remain in centers of specialism or pilot programs, rather than routinely or equitably integrated into clinical or athletic practice. This narrative review synthesizes translational evidence across the life course—in pregnancy, paediatrics, adult medicine, geriatrics, and sportomics—to find reproducible clinical and performance examples which enable precision-based alternative approaches to management, outcome, or preparation; and to reshape those examples into pragmatic, scalable priorities which minimize inequity, and maximize benefit. We undertook a structured narrative synthesis of peer-reviewed literature, trials, clinician translation programs, implementation studies, and sportomics reports, prioritizing examples that demonstrate utility, reproducibility, and impact. Important findings suggest that multi-omics and rapid sequencing improve diagnostic yield and time to diagnosis. Molecular profiling and circulating tumor DNA help realize adaptive treatment selection. Integrated genomics, metabolomics, wearable physiology, and AI analytics facilitate individualized training, injury-risk stratification, and recovery optimization. But systematic value is limited by insufficient representative validation, dataset bias, poor interoperability, regulatory uncertainty, workforce preparedness, and inequities of access. Converting a promise into population- and performance-level value requires coordinated action across four fronts: representative validation; interoperable, privacy-preserving infrastructures; clinician- and coach-centered implementation; and templates for scalable, cost-sensitive deployment.

## 1. Introduction

Personalized medicine, also called precision medicine, is transforming healthcare by tailoring prevention, diagnosis, and treatment strategies to individual patient characteristics, including genetic, molecular, and environmental factors [1,2]. By integrating multi-omics approaches such as genomics, proteomics, and metabolomics, it allows a deeper understanding of disease susceptibility and variability in therapeutic response [3,4,5]. This approach moves beyond the traditional “one-size-fits-all” model, aiming to enhance clinical efficacy and reduce adverse drug reactions [1,6,7].

Clinical applications in oncology have advanced significantly, with neoantigen-specific T cell therapies and personalized cancer vaccines demonstrating efficacy in metastatic melanoma and lymphoplasmacytic lymphoma [8].

Pharmacogenomic strategies have shown promise in autoimmune and inflammatory diseases by optimizing biologic drug response and minimizing treatment failure [9,10,11]. In cardiovascular medicine, AI-assisted personalized interventions improve outcomes in conditions such as atrial fibrillation and hypertension [12,13,14].

In obstetrics, personalized approaches allow precise risk prediction for gestational diabetes, preeclampsia, and spontaneous preterm birth, enhancing maternal and neonatal outcomes [15,16]. Nutrigenomic interventions guided by microbiota profiling improve cardiometabolic health and reduce [17,18,19]. Pediatric precision medicine, particularly genomics-driven therapies, offers improved management of solid tumors and bronchiectasis in children [19,20,21].

Geriatric populations benefit from comprehensive assessments combined with personalized interventions, optimizing therapy in older adults [22,23,24]. Metabolomics and proteomics approaches enhance disease stratification and early detection in gastric and colorectal cancers [5,25,26]. AI applications in drug discovery, delivery, and predictive diagnostics accelerate the development of individualized treatment plans [27,28,29].

Personalized interventions are increasingly applied to chronic diseases, including diabetes, obesity, asthma, and urticaria, addressing patient heterogeneity in therapeutic response [30,31,32]. The integration of AI and digital health platforms supports early detection, risk stratification, and precision screening [27,33,34]. Despite its potential, personalized medicine faces challenges including high costs, data privacy, limited accessibility, and integration into routine clinical practice as shown in Figure 1 [2,35,36].

Overall, personalized medicine is shaping a predictive, preventive, personalized, and participatory future in healthcare, leveraging patient-specific data to optimize outcomes, reduce healthcare costs, and improve quality of life globally [16,17,29].

## 2. Methodology

This narrative review was conducted using a structured and systematic literature search to identify and synthesize evidence on precision and personalized medicine across the life course, including pregnancy, pediatrics, adulthood, geriatrics, and athletic performance (sportomics). Peer-reviewed publications were retrieved from PubMed/MEDLINE, Scopus, Web of Science, Embase, and the Cochrane Library, with supplementary screening of Google Scholar, reference lists of key articles, and registered clinical trials to ensure comprehensive coverage. The search period spanned 2019 to 2025, capturing the era of rapid integration of multi-omics technologies, artificial intelligence, and digital health tools into translational and clinical research. Search terms were applied using controlled vocabulary and free-text keywords, either alone or in Boolean combinations, and included: precision medicine, personalized medicine, multi-omics, genomics, proteomics, metabolomics, pharmacogenomics, artificial intelligence, machine learning, wearable sensors, pregnancy, pediatrics, adult disease, geriatrics, oncology, cardiovascular disease, neurodegenerative disorders, asthma, diabetes, sportomics, and pharmacoeconomics. Eligible studies were original peer-reviewed research articles, randomized controlled trials, observational and implementation studies, systematic reviews, and high-quality narrative reviews that reported clinical, translational, performance, or economic outcomes related to individualized care or training strategies in human populations. Articles not published in English, non-peer-reviewed sources, and preclinical-only animal studies were excluded unless directly linked to human translation. Study selection was performed through title/abstract screening followed by full-text review, and relevant data were extracted on population, precision modality (omics, AI, pharmacogenomics, or digital tools), outcomes, and implementation considerations. Owing to methodological heterogeneity across studies, a meta-analysis was not pursued; instead, findings were synthesized using a structured narrative approach, organizing evidence by life stage and thematic domain to identify reproducible applications, translational impact, and systemic barriers to equitable implementation.

## 3. Clinical Pharmacology, Pharmacokinetics, and the Role of Therapeutic Drug Monitoring in Personalized Medicine

### 3.1. Clinical Pharmacology and Therapeutic Drug Monitoring

Clinical pharmacology is a specialized field within pharmacology that concentrates on studying the effects of drugs in humans, including their mechanisms of action, therapeutic benefits, adverse reactions, and interactions. Its goal is to enhance drug therapy by recognizing individual patient characteristics such as age, genetic factors, organ functionality, and other medications that impact drug responses [37]. One critical aspect of clinical pharmacology is pharmacokinetics, which explains how the body absorbs, distributes, metabolizes, and eliminates medications. In conjunction with this, pharmacodynamics investigates how drugs achieve their therapeutic and side effects, as well as the correlation between drug concentration and clinical outcomes. Therapeutic Drug Monitoring (TDM) is a vital practice in clinical pharmacology that involves measuring specific drug levels in biological samples, typically blood, to inform dosage adjustments. TDM is especially crucial for drugs with narrow therapeutic ranges, considerable interindividual variability, or high levels of toxicity, including immunosuppressants, antiepileptics, and specific chemotherapy agents [38]. The use of TDM enables healthcare providers to tailor treatment by modifying doses based on observed drug concentrations and specific patient characteristics. This strategy improves treatment effectiveness while reducing the risk of adverse effects, drug toxicity, and therapeutic failure [39]. Pharmacogenomic markers listed in Table 1 highlight the importance of genotype-guided therapy, which is further complemented by clinical pharmacology and TDM.

### 3.2. Mid-Pregnancy Monitoring and Placental Biomarkers

Mid-pregnancy, maternal and placental biomarkers inform on fetal development and maternal adaptations that are happening over these critical later gestational weeks. At this point in time placental function is the most important prognostic influence both for maternal and fetal outcomes. Placental Growth Factor (PlGF) continues to rise, reaching a peak, and optimal placental angiogenesis, around mid-gestation. PlGF consistently low during mid-gestation is demonstrated to have strong associations with preeclampsia and fetal growth restriction (FGR). Alongside placental growth, soluble fms-like tyrosine kinase-1 (sFlt-1) rises too; an adaptive response to rising PlGF towards and during mid-gestation is an increasing level of sFlt-1 compared to weakly increasing PlGF, which indicates an anti-angiogenic state and is often pre-clinical evidence of other placental biomarkers relating to other pregnancy complications like preeclampsia and also have been shown to demonstrate placental insufficiency. Other soluble markers to indicate placental dysfunction, like soluble endoglin (sEng), demonstrate modest elevations mid-pregnancy as relates to sFlt-1, these too represent elements of ‘endothelial dysfunction’ High levels of both sEng and sFlt-1 reliably represent worse adverse events in severe preeclampsia and catastrophic neonatal outcomes, thus proving additional prognostic value of these common soluble inflammatory indexes of placental dysfunction [45].

Mid-pregnancy imaging and the incorporation of biomarkers better refine and personalize screening assessments. Doppler of uterine arteries provides a helpful evaluation of placental blood flow through well-defined ‘abnormal’ findings. The combination of these studies and biomarker levels of PlGF and sFlt-1 allows better prediction of adverse downstream effects, such as intrauterine growth restriction (IUGR). Advances in placental transcriptomics and proteomics have also identified gene expression signatures for genetic programs associated with angiogenesis, oxidative stress, and immune tolerance. Targeted profiling such as this may detect high-risk pregnancies even before clinical symptoms manifest and provide the rationale for early intervention or other pre-emptive management. Applications of artificial intelligence (AI) for mid-pregnancy monitoring are emerging as viable and powerful alternatives.

Artificial intelligence (AI) or machine learning models can incorporate maternal serum biomarkers, placental imaging, and demographic/clinical factors to predict adverse outcomes such as preeclampsia, preterm birth, and fetal growth restriction with greater accuracy than conventional risk scoring systems. Personalized surveillance schedules and intervention approaches can be designed for each pregnancy using AI-enabled predictive models [46,47,48,49].

### 3.3. Late Pregnancy and Fetal Biomarkers

In the late pregnancy period, fetal well-being is the principal focus. Surveillance of growth is important as preparation for delivery advances. Maternal, placental, and fetal biomarkers need to be integrated to anticipate complications and plan interventions at this stage. Notably, nuchal translucency (NT) measured in the first and second trimesters remains important at this stage of pregnancy, where increased NT is associated with chromosomal problems, including trisomy 21, trisomy 18, and other syndromes. Alpha-fetoprotein (AFP) in maternal serum, assessed during the second trimester, is used to screen for neural tube defects (NTDs). High AFP allows for quick diagnostic imaging, and when deemed necessary, interventions are often possible and timely. Cytokines and other inflammatory markers measured in fetal cord blood reflect fetal immune development. When levels are elevated, clinicians are aware of an increased morbidity risk, which may affect their decisions on whether to intervene and/or deliver early.

Our understanding of maternal environment and maternal metabolic status on the methylation of fetal DNA suggests novel long-term health risks ranging from cardiometabolic disease, asthma, and neurodevelopmental disorders. Precision-guided interventions, including targeted maternal diet and supplementation to modify methylation, may lessen the burden associated with environmental factors across the maternal/fetal continuum [50,51,52].

#### Fetal-Specific Therapeutic Interventions

Gene therapy in utero: Preclinical studies indicate that it may be possible to repair congenital genetic defects before birth by exploiting fetal immune tolerance and reducing or eliminating the risk of rejection.

Stem cell transplant for hemoglobinopathies and immunodeficiencies can be performed before birth to allow donor cells to engraft and protect the organ. Enzyme replacement therapy (ERT): Trials in lysosomal storage disorders suggest intrauterine administration can prevent irreversible organ damage [53].

Enzyme replacement therapy (ERT)—trials in lysosomal storage disorders show that in utero treatment can maintain organ function before irreversible damage occurs.

Minimally invasive fetal surgery for a variety of conditions—spina bifida, congenital diaphragmatic hernia, etc., has improved outcomes and safety through precise imaging and robotic-assisted techniques, and the integration of fetal optimal biomarkers with maternal and placental information will facilitate the creation of multivariate predictive models that incorporate artificial intelligence algorithms to enhance our understanding of preterm birth, preeclampsia, and fetal growth restriction, and inform an individualized monitoring schedule and timing of effective interventions. The concept of digital twins of pregnancy employing omics, imaging, physiological, and biomarker data has positioned us at the leading edge of proactive obstetric care [54,55,56].

### 3.4. Personalized Interventions and Clinical Applications

Personalized medicine does not stop at identifying maternal-fetal risk but focuses on performing effective and evidence-based interventions relative to each mother’s unique profile. A combination of maternal, placental, and fetal biomarkers will allow clinicians to identify appropriate stage-specific strategies for optimizing outcomes for both the mother and her neonate.

Early Pregnancy Interventions: women identified as being at high-risk based on biomarker profiles (PAPP-A, β-hCG, PlGF, sFlt-1, and immune biomarkers) can be treated at the earliest opportunity. For example, initiation of low-dose aspirin in the first trimester significantly decreases the risk of preeclampsia in women with abnormal angiogenic biomarker levels. Timely treatment of hypertension using appropriate antihypertensive medications optimizes maternal and fetal health [52].

## 4. Pediatric Precision Medicine

Instead of treating everyone the same way, this method of treating and preventing diseases looks at each person’s genes, environment, and lifestyle. Doctors typically base medicine dosages for children only on weight, or occasionally gestational age, which is the number of weeks since the first day of the mother’s last menstrual period until the baby is born. However, this overlooks other significant factors, such as genetic variations that may alter a drug’s efficacy or safety. Now we know more about how a child’s development (ontogeny) and genes affect drug responses. This gives us new chances to apply precision medicine to all stages [53]. Through proactive illness prevention and health preservation, precision medicine offers a new direction away from traditional, reactive disease control. We can forecast health patterns, identify an individual’s short- and long-term illness risks at the molecular level, and put preventive measures in place that are specific to that individual based on their genomic profile and environmental effects. We can create biomarkers for early disease identification, track the course of diseases, and create innovative, tailored treatments that may stop the progression of diseases and restore health. By choosing treatments based on a person’s molecular profile, we might be able to maximize benefits and minimize side effects [54].

### 4.1. Pediatric Pharmacogenomics and Genome Sequencing

Genetics has long been recognized as important in pediatrics. At the start of the 20th century, Sir Archibald Garrott highlighted the role of heredity in the variability of drug action. Subsequent discoveries of polymorphisms in drug-metabolizing enzymes moved this field into clinical practice. Today, more than 260 FDA-approved drugs carry pharmacogenomic information, with testing being routine for therapies such as warfarin. Recent advances in genome sequencing have further expanded precision medicine. Unlike targeted gene panels, whole-genome sequencing can detect nearly all DNA variation and has diagnostic potential for more than 7000 known genetic diseases (e.g., cystic fibrosis, Duchenne muscular dystrophy, familial hypercholesterolemia, hemophilia). It is also increasingly applied to conditions with both rare and common presentations (e.g., autism spectrum disorder, cardiomyopathy, epilepsy, cancer). Genome sequencing follows a structured process: Clinical evaluation, which focuses on collecting phenotype and family history; Genomic analysis, which involves laboratory detection of variants; and Interpretation, which links genetic findings to clinical manifestations. Some laboratories additionally report secondary findings relevant to actionable conditions beyond the primary reason for testing [53]. Pharmacogenomics is crucial in precision medicine because it explains how a person’s genetic differences affect how drugs are broken down, how well they work, and whether they cause side effects. People respond differently to the drug due to genetic variations that change pharmacokinetics, which shows what the body does to the drug (how it absorbs, distributes, metabolizes, and eliminates it), and pharmacodynamics, which elucidates what the drug does to the body (how it interacts with receptors, enzymes, and transporters). By studying these genetic variations, doctors can determine drug choice or dose to match each patient’s genetics. This enhances treatment results and reduces harmful side effects. The metabolism of drugs relies on the cytochrome P450 (CYP) enzyme family (a family of liver enzymes that break down most drugs), phase II conjugation enzymes (enzymes that make drugs more water-soluble to be excreted), and membrane-bound transporters (proteins that move drugs in and out of cells). These enzymes and transporters control how drugs are chemically altered (biotransformed) and cleared from the body. This determines how much of the drug stays available in the blood and how effective it is. Each person’s drug metabolism is affected by their haplotypes (a group of genetic variants that tend to be inherited together). Populations share common haplotypes, which shape how drugs are processed. Variations (polymorphisms) in these genes can cause different metabolic categories like ultrarapid metabolism (UM), extensive metabolism (EM), intermediate metabolism (IM), or poor metabolism (PM). By studying these variations, clinicians can personalize drug treatment, making it safer and more effective [53,55].

### 4.2. Pediatric Precision Medicine in Cancer

Childhood cancers affect about 400,000 kids a year worldwide. The International Classification of Childhood Cancer (ICCC) is a system used by scientists and doctors to group childhood cancers into categories: leukemias, lymphomas, and brain tumors. Most are rare. Before chemotherapy, survival was less than 25%, but since the 1960s, survival has dramatically improved thanks to new treatments. Recent genomic research shows that children’s cancers are molecularly different from adult cancers, which means they need different treatment approaches. In the last 20 years, advances in genetics and technology have made a huge difference. The Human Genome Project (completed in 2003) mapped human DNA. Next-Generation Sequencing (NGS) allows scientists to read DNA much faster and cheaper, and Omics sciences (like genomics, transcriptomics, proteomics, metabolomics) let researchers study all genes, proteins, and metabolites at once. Next-Generation Sequencing (NGS) and related “omics” technologies enable clinicians to read the genetic and molecular makeup of a child’s tumor in great depth. By studying the mutational profiles of tumors, doctors can better diagnose, classify risk, and predict prognosis. Scientists often use a stepwise approach: First, targeted NGS (focuses on a set of known cancer genes). If there are no results, Whole-Exome Sequencing (WES) or Whole-Genome Sequencing (WGS). This allows them to detect amplifications, point mutations, and gene fusions, which are often disease-specific. Pediatric cancers generally have fewer mutations than adult cancers. Still, NGS has revealed new driver genes (genes that push cancer growth). Sometimes, in aggressive cancers, no single “driver” gene can be identified. Some tumors (e.g., hepatoblastoma in very young children, rhabdoid tumors) have very few mutations but may exhibit recurrent alterations in a single gene. Others (some high-grade gliomas or aggressive tumors) may harbor many changes or complex structural alterations [56].

### 4.3. Pediatric Precision Medicine in Neurodegenerative Diseases

Neurodegenerative diseases, including Alzheimer’s disease (AD), Parkinson’s disease (PD), Frontotemporal lobar Degeneration (FTLD), and Amyotrophic Lateral Sclerosis (ALS), have common features, such as the loss of neurons and synapses, the presence of chronic inflammation in the brain (neuroinflammation), and the buildup of abnormal protein aggregates inside or around brain cells. Furthermore, recent studies of human postmortem brains have shown that many patients do not have just one type of abnormal protein (“proteinopathy”) but often show multiple protein pathologies occurring together. These combinations of abnormal proteins can worsen or change the course of the disease. Despite this, most current research and drug development efforts focus solely on single-protein problems, such as amyloid-beta (Aβ) in Alzheimer’s disease, tau in tauopathies, or alpha-synuclein in Parkinson’s disease. Unfortunately, this single-target approach has had only limited success [57]. Therefore, personalized medicine is applied by considering genetic, environmental, and lifestyle factors. Recently, many scientists have tried to apply precision medicine to their patients. In recent years, precision medicine for neurodevelopmental disorders has mostly focused on using genomic sequencing technologies. These tools help scientists find mutations in single genes that cause rare disorders, allowing for early diagnosis and a better understanding of how the disease develops. This field has been successful in some cases. For instance, spinal muscular atrophy (SMA) is considered a model case. Mutations in single genes are the cause of SMA, and once that was discovered, researchers developed genetic therapies that directly address the defect. When these therapies are given to babies before symptoms appear, they can prevent the severe motor decline that used to be inevitable. Many children who would have lost the ability to walk can now grow up walking independently. Because of this breakthrough, newborn screening for SMA has been introduced, making early diagnosis and treatment more widely available. This example shows how powerful genomic-based precision medicine can be in rare, single-gene conditions.

Charcot-Marie Tooth disease, muscular dystrophies, and congenital myopathies are among the other neuromuscular disorders that are undergoing similar developments. To improve function or slow the development of disease, clinical trials are currently examining medicines that target different biochemical pathways. However, the story becomes more complicated with other neurodevelopmental disorders (NDDs). Scientists have found that over 900 different genes can play a role in brain development and function. The “diagnostic yield” of sequencing, meaning how often genetic testing identifies the cause, varies depending on the disorder. For intellectual disability, sequencing can now explain over 40% of cases, which is a major improvement. But for more common NDDs like autism, ADHD, OCD, or Tourette’s syndrome, the yield is much lower. In these conditions, the causes are not usually due to a single defective gene. Instead, they include many genes, each contributing to a small effect, combined with environmental influences. Even when doctors do find a genetic diagnosis, this does not always translate into a treatment. Nowadays, focused therapies (such as vitamin/cofactor supplements, enzyme treatment, or gene substitution) only assist less than 10% of children with a proven genetic diagnosis. For the majority, genetic diagnosis mainly helps with genetic counseling, meaning families can understand recurrence risks, make reproductive decisions, or plan for the future.

This limitation highlights a major issue: while single-gene discoveries are important, they only solve a small part of the problem. Most children with NDDs do not have a “fixable” single-gene mutation; hence, gene therapy will not be a solution for them even in the future. Their conditions are shaped by the interaction of many genes and non-genetic factors [58].

### 4.4. Pediatric Precision Medicine in Asthma

Pediatric asthma remains one of the most common chronic diseases in childhood, characterized by marked heterogeneity in clinical presentation, inflammatory pathways, and response to therapy. Recent advances strongly support the integration of precision medicine into pediatric asthma management [59].

Genetic makeup plays a central role in shaping disease susceptibility, severity, and therapeutic responsiveness. Notably, a genome-wide study revealed an interaction between CDHR3 and GSDMB variants associated with early-onset severe asthma in children, a finding that underscores the importance of considering specific genetic backgrounds when stratifying risk and tailoring treatment [60].

Environmental exposures such as air pollution, allergens, viral infections, and secondhand smoke interact with genetic predispositions to determine asthma onset and progression in childhood. This gene environment interplay supports a shift from single-factor risk models to integrated “exposome + genomics” frameworks for disease prediction and prevention [61].

Beyond genetics, epigenetic mechanisms (e.g., DNA methylation, miRNA regulation, histone modifications) have emerged as potential modulators of asthma susceptibility and phenotypes in children. These epigenetic markers offer promising avenues for early diagnosis, prognostic assessment, and development of personalized interventions [62].

### 4.5. Pharmacogenomics in Pediatric Cardiovascular Therapy

Although most research in cardiovascular pharmacogenomics focuses on adults, recent pediatric reviews highlight that similar genetic mechanisms influence drug response in children [63].

Variants in CYP2C9 and VKORC1 are known to affect warfarin sensitivity, and although pediatric-specific trials remain limited, available evidence suggests that reduced-function alleles may lower dose requirements and increase bleeding risk [64].

Similarly, genetic variation in CYP2C19 may influence clopidogrel activation in children with congenital heart disease, suggesting that genotype-guided antiplatelet therapy could improve outcomes [65].

Together, these observations highlight the growing potential of pharmacogenomic-guided dosing to enhance both safety and therapeutic effectiveness in pediatric cardiology [66].

### 4.6. Antiepileptic Drug (AED) Response

Pharmacogenomics plays a well-established role in antiepileptic drug response, with several associations confirmed in pediatric populations [67].

Carbamazepine hypersensitivity, including severe reactions such as Stevens–Johnson syndrome, is strongly linked to HLA-B*15:02 and warrants genetic screening in at-risk groups [68].
(A)Phenytoin metabolism is significantly influenced by CYP2C9 and CYP2C19 variants, which alter serum levels and toxicity risk in children whose metabolic pathways are still developing [69].(B)Valproic acid hepatotoxicity has been associated with POLG mutations, making pre-treatment screening essential to avoid severe mitochondrial complications [70].

Collectively, these findings demonstrate how pharmacogenomics enhances safety, reduces adverse reactions, and supports individualized treatment strategies in pediatric epilepsy.

## 5. Personalized Medicine in Adults

In the last five years, personalized medicine for adults has moved from an aspirational concept to a practical approach that measurably influences clinical choices and outcomes. Instead of depending on population-level averages, leading initiatives now begin with precise individual assessments at the molecular level (genome, transcriptome, neo-antigens), physiological level (hemodynamics, electrophysiology), and behavioral level (nutrition, session-by-session engagement). Therapies, devices, and care pathways are then adjusted to maximize benefit while reducing risk. Evidence from randomized and prospective trials in cardiology, oncology, transplantation, metabolic disorders, perioperative medicine, mental health, and digital musculoskeletal care shows that personalization is most effective when it is actionable (changing clinician decisions), adaptive (evolving with new information), and benchmarked against rigorous comparators [71].

### 5.1. Advances in Personalized Oncology

#### 5.1.1. Radiotheranostics

In patients with somatostatin receptor–positive neuroendocrine tumors, combining ^^^131I-MIBG with dosimetry-guided ^^^90Y-DOTATOC peptide receptor radionuclide therapy (PRRT) improved tumor targeting while maintaining safety [72].

#### 5.1.2. Neoantigen-Based Therapies

In early-phase trials, personalized neoantigen vaccines combined with anti–PD-1 therapy elicited tumor-specific T-cell responses in advanced melanoma, NSCLC, and bladder cancer, with accurate HLA matching proving critical. More recently, a 2025 Nature study in renal cell carcinoma demonstrated that individualized vaccine payloads can induce antitumor immunity, while large studies such as I-PREDICT and WINTHER confirmed that integrating genomic and transcriptomic profiling broadens treatment options beyond anatomical staging [73,74]

#### 5.1.3. Pharmacogenomics in AML

Composite scoring models enable stratification of older patients by weighing benefit versus toxicity, moving beyond mutational catalogs toward actionable decision tools. Expanding this approach, the Beat AML Master Trial demonstrated the feasibility of rapid genomic profiling to direct therapy within days of diagnosis in a real-world, multi-center setting [75].

### 5.2. Metabolic Disease and Personalized Nutrition

Dietary recommendations tailored using microbiome profiles, post-prandial glycemic and lipemic responses, and prior dietary history led to markedly greater improvements in weight loss, lipid parameters, and overall diet quality compared with standard generalized advice. These findings highlight the potential of combining multi-omics baseline profiling with adaptive feedback to optimize nutritional interventions [16].

The PDM-ProValue program version 2.1 demonstrated that combining integrated digital tools with structured feedback in diabetes care leads to clinically significant reductions in HbA1c. This effect stems from enabling personalized, continuous treatment adjustments rather than relying solely on infrequent in-clinic titration [76].

Adult weight-loss interventions that combine web-based behavior change strategies with dietitian-led e-coaching outperform generic online content, largely by enhancing adherence and supporting sustained lifestyle modifications [77].

### 5.3. Personalized Hemodynamic Management in Perioperative Care

In a pilot bicentric randomized controlled trial involving patients undergoing major non-cardiac surgery, intraoperative blood pressure was managed according to each patient’s preoperative nighttime mean arterial pressure (MAP) instead of relying on the conventional fixed threshold of approximately 65 mmHg. The findings revealed that personalized MAP targets often diverged from the standard cutoff, proved feasible to apply in practice, and may improve the precision of tissue perfusion, especially in older or high-risk individuals [77].

### 5.4. Precision Approaches in Mental Health

Advances in precision mental health are becoming increasingly refined. In a trial focusing on late-life depression with coexisting hearing impairment, a predictive tool was employed to identify which patients were most likely to benefit from hearing aids by integrating both sensory data and depressive symptom profiles. This approach enables tailored adjunctive interventions for a population that is often underrepresented in research and clinical practice [78].

A network-informed approach to treating eating disorders, which sequences therapy modules based on each patient’s individual symptom network (identifying symptoms that sustain others), was found to be both feasible and acceptable when compared with enhanced CBT. However, further research is needed to clarify its impact on long-term outcomes [79].

Precision psychiatry now reaches beyond network-informed approaches in eating disorders. The EMBARC trial established predictive markers that differentiate antidepressant from placebo response, showing that treatment decisions can be improved by combining cognitive, clinical, and biological moderators. Likewise, the Personalized Advantage Index demonstrated that individualized models can forecast whether cognitive therapy or interpersonal therapy is more likely to provide lasting benefit in depression [80].

### 5.5. Cross-Cutting Insights into Personalized Medicine

Several cross-cutting studies highlight design features that drive successful personalization.

Heterogeneity measurement: The TAILORED-AF trial showed particularly pronounced benefit in patients with longer atrial fibrillation duration and greater structural remodeling, precisely the group in which anatomical pulmonary vein isolation (PVI) has traditionally yielded poor outcomes [12]. Adaptive Algorithms and Feedback: Phenotype-guided tacrolimus dosing dynamically adjusts with ongoing drug level monitoring, allowing therapy to evolve over time [81].

Multifactorial models (beyond single biomarkers): A kidney transplant study on tacrolimus exposure demonstrated the value of integrating genetics, laboratory measures, and physiological parameters to explain variability and guide dosing [74].

### 5.6. Digital Personalization in Musculoskeletal Care

A recent trial in remote management of low-back pain employed a machine-learning model to predict, session by session, which adults were likely to achieve clinically meaningful pain relief. By identifying probable non-responders early, the system enabled tailored adjustments in exercise intensity and adjunctive therapies, illustrating a “learn-as-you-treat” approach to personalization [82].

### 5.7. Personalized Nutrition and Diabetes Management

Beyond individual trials, systematic evaluations show that microbiome-informed dietary interventions consistently outperform generalized diet plans in improving weight, glycemic control, and lipid profiles. The PDM-ProValue program demonstrated sustained HbA1c reductions through app-based monitoring and structured feedback, while microbiota-focused RCTs reported that tailored diets improved hyperglycemia, hypertension, and inflammatory markers [83].

### 5.8. Personalized Approaches in Neurology and Neurodegeneration

Personalized strategies are increasingly shaping neurology. In Parkinson’s disease, a randomized cross-over trial of individualized transcranial alternating current stimulation (tACS) combined with physiotherapy enhanced both motor and cognitive outcomes, demonstrating the feasibility of tailoring stimulation frequency to each patient’s electrophysiological profile. In Alzheimer’s disease, a precision NSAID therapy model aligned drug use with patient-specific risk profiles, providing a framework for stratified approaches to neurodegenerative care [84].

## 6. Personalized Medicine in Geriatric Patients (Older than 60 Years Old)

Personalized medicine has improved healthcare for age-related diseases in the geriatric population through genetic profiling.

Personalized medicine has applications for age-related diseases such as Alzheimer's Disease (AD), Parkinson's Disease (PD), and depression.

First: Personalized Therapies in Neurodegenerative Diseases

In Alzheimer’s disease (AD), personalized medicine helps predict risk and guide targeted treatments through APOE ε4 and other genetic variants, while in Parkinson’s Disease (PD), mutations in the GBA and LRRK2 genes influence disease course and therapy response.

Second: Personalized Therapies in Psychiatric Conditions.

### 6.1. Anxiety Disorders

CYP enzyme genetic differences can affect how benzodiazepines and other frequently prescribed anti-anxiety medications are metabolized in treating anxiety disorders, including generalized anxiety disorder (GAD) and panic disorder. Also, neurotransmitter-related genetic markers, including the catechol-O-methyltransferase (COMT) and serotonin transporter (SLC6A4) genes, have been studied as potential targets for individualized anxiety treatment [85].

### 6.2. Depression

Geriatric Depressive Disorder is classified according to the Diagnostic and Statistical Manual of Mental Disorders, 5th Edition (DSM-5) into major depressive disorder (MDD), minor depressive disorder (MnDD), dysthymic disorder, bipolar I disorder, and adjustment disorder with depressive mood. Many patients still do not experience remission.

Personalized medicine (PM) is based on the understanding that patients with the same condition differ from one another. This method states that the objective is to customize suitable treatments for certain patient populations. PM makes it possible to forecast therapy’s efficacy for a certain patient [86].

In addition to Psychiatric Conditions and Neurodegenerative Diseases, Leukemia, especially AML (Acute Myeloid Leukemia), challenges in elderly patients are worse than in younger patients.

Contributing factors include higher rates of high-risk cytogenetic and molecular abnormalities, more secondary or therapy-related AM comorbidities that restrict intensive therapy or transplantation. In personalized medicine, each patient’s pretreatment bone marrow or blood undergoes high-throughput sequencing (HTS), cytogenetic testing, and specific assays, such as FLT3-ITD analysis. These provide a molecular “fingerprint” of the leukemia, identifying mutations, cytogenetic abnormalities, and their variant allele frequencies (VAF) [75].

## 7. Pharmacoeconomic Perspectives on Drug Development and Rational Therapeutic Choices

Pharmacoeconomics is a subfield of health economics that examines, quantifies, and contrasts the costs and effects of pharmaceutical goods and services. It facilitates the development of an economic relationship that encompasses medication research, manufacturing, delivery, storage, cost, and subsequent human use. When comparing two medications in the same therapeutic class, Pharmacoeconomics can be a huge aid in decision-making when determining the cost and availability of the appropriate treatment to the appropriate patient at the appropriate time. Therefore, decision-makers need to judge if the new intervention is affordable and makes the best use of available resources [87].

The economic evaluation of healthcare interventions, particularly pharmaceutical treatments, relies on comparing different options based on their costs and resulting health outcomes. This approach helps determine which therapy provides the greatest value. The main types of pharmacoeconomic analyses include cost-minimization, cost-effectiveness, cost-utility, and cost–benefit analyses. Cost-minimization analysis is used when two or more drug options produce identical therapeutic outcomes. Since their clinical effectiveness is the same, the comparison focuses solely on cost differences. In this case, the preferred choice is the drug that provides the same health benefit at a lower cost. Cost-effectiveness analysis (CEA) is used to compare two or more treatment options in terms of their costs and clinical outcomes. Health outcomes must be expressed in the same measurable units, allowing fair comparison. When one treatment achieves better results at a higher cost, CEA helps determine whether the additional benefit is worth the extra cost. CBA is an analytical method used to compare two drugs, requiring that both costs and benefits be expressed in monetary terms. This means the method must assign a monetary value to the health effects of each drug. However, CBA is not commonly used and raises ethical concerns because it involves converting health outcomes into money, which is difficult and controversial. And the last is Cost–Utility Analysis (CUA) is a type of economic evaluation used to compare two or more healthcare treatments by considering both their costs and their impact on the quality and length of life in terms of Quality-Adjusted Life Years (QALYs), a measure that combines quantity of life (how long a person lives) and quality of life (how healthy or comfortable those years are). The main goal of CUA is to identify which treatment provides the most health value for the cost spent, considering both life extension and life quality [88].

Due to the assessment of whether the benefits justify the limitations, pharmacoeconomics is crucial for determining if tailored therapies are cost-effective. However, many previous evaluations have shown contradictory evidence about the economic usefulness of PM due to a lack of clinical data, variable methodology among analyses, and low study quality. Despite these challenges, recent studies show that pharmacoeconomic evaluations in PM are now more accurate, suggesting that efforts to identify the factors influencing cost-effectiveness have increased. Thus, pharmacoeconomics provides the foundation for evaluating the financial viability of customized treatments and enables a more seamless integration of PM into healthcare systems as shown in Figure 2 [89].

## 8. Artificial Intelligence as the Backbone of Personalized Medicine

Artificial intelligence (AI) is redefining personalized medicine by transforming multi-layered data into actionable treatment decisions, allowing care to shift from population averages to precise individual targets [90]. The WINTHER trial highlighted how transcriptomic and genomic data integrated with decision support expanded the scope of therapy matching in oncology, providing a real-world example of AI-enabled precision care [91].

### 8.1. AI in Oncology: From Multi-Omics to Neoantigen Discovery

Deep molecular profiling of synovial biopsies in the STRAP trial illustrates how multi-omic analysis can predict biologic therapy response in rheumatoid arthritis, with direct implications for AI models in cancer [71]. The I-PREDICT study demonstrated how molecular profiling enables AI-assisted combination therapy matching in advanced cancers [92]. The PNOC003 pediatric trial showed the feasibility of AI-guided personalized therapy in diffuse intrinsic pontine glioma, where genomic signatures informed targeted interventions [93]. Neoantigen vaccine studies illustrate AI’s role in antigen prioritization, as seen in melanoma and NSCLC, where personalized immunotherapy combined with PD-1 blockade achieved promising efficacy [94]. A first-in-human renal carcinoma vaccine confirmed robust immune induction, demonstrating how AI-driven neoantigen mapping supports translational oncology [94].

Landscape analysis of follicular lymphoma identified feasible personalized vaccine strategies, which AI can accelerate by modeling peptide stability and presentation [93]. Adoptive autologous neoantigen-specific T cell therapy in melanoma further demonstrated that computational pipelines enhance patient-specific target identification [95]. ctDNA monitoring provides predictive recurrence signatures in colon cancer, offering a clear domain for AI integration in risk stratification [95].

### 8.2. AI in Cardiovascular Medicine: Predictive Models for Risk and Therapy Optimization

Cardiovascular MRI validated idiopathic dilated cardiomyopathy with precision, illustrating AI’s ability to merge imaging and genetic data for accurate subtyping [96]. Rare variant analysis further revealed genetic determinants of severity, strengthening AI models for prognosis [96]. Insertable cardiac monitors combined with AI-derived risk scores enabled early personalized interventions in heart failure [97]. Dynamic prediction of chronic heart failure using 92 biomarkers showed how longitudinal datasets enrich AI performance [85].

Filtered forecasting models predicted glaucoma progression and guided intraocular pressure targets, reflecting AI’s capacity for individualized therapy [98]. A randomized trial of four antihypertensives revealed patient-level heterogeneity, underscoring the need for AI-driven drug selection strategies [98].

### 8.3. AI in Pulmonology and Infectious Diseases: Phenotypes, Biomarkers, and Adaptive Therapies

The “treatable traits” approach in asthma demonstrates how biomarker-guided personalization benefits from AI classifiers of phenotypes [99]. Childhood asthma RCTs show that individualized therapy is clinically feasible, offering real-world data for AI-based decision aids [99]. Mechanistic mapping of JAK1 inhibition in asthma created datasets ideal for AI algorithms predicting responder subsets [88]. Gene expression scoring in vasodilatory shock showed that steroid therapy can be restricted to biologically defined subgroups, a principle expandable through AI [100]. Machine learning rules identified which children with watery diarrhea benefited from azithromycin, demonstrating AI-enabled antimicrobial stewardship [100].

### 8.4. AI in Neurology and Psychiatry: Cognitive Markers, Digital Tools, and Personalized Interventions

The Personalized Advantage Index predicted long-term depression outcomes by selecting optimal therapy type, exemplifying AI’s role in psychotherapy matching [101]. Precision tools identified who benefits most from hearing aids in late-life depression, integrating sensory interventions with AI-informed psychiatry [102]. Cognitive markers for ADHD neurofeedback were successfully modeled, supporting computational psychiatry as a foundation for AI-based treatment prediction [102]. SMART trial designs allowed adaptive sequencing for mothers with ADHD, a structure directly synergistic with AI-based reinforcement learning [103].

Personalized tACS with physiotherapy in Parkinson’s disease confirmed symptom improvement, paving the way for AI-tuned neuromodulation [104]. Internet-based conversational engagement trials demonstrated that AI could help identify which patients gain the most from digital social interventions against cognitive decline [105].

### 8.5. AI in Metabolism and Nutrition: Microbiome, Dietomics, and Digital Twins

Personalized nutrition programs improved cardiometabolic health, providing scalable data for AI-enhanced precision nutrition [106]. Microbiota-based personalization improved glycemia and reduced inflammation, demonstrating AI’s role in interpreting microbiome signals.

The Food4Me trial confirmed that personalized advice, even without extreme diets, can alter health behavior at scale [107]. Twelve-week personalized dietary interventions improved vascular function and lowered cardiovascular risk markers, offering datasets for AI-powered diet digital twins [108]. Intensive dietary personalization in colorectal cancer altered base-excision repair activity, demonstrating how AI could map nutritional interventions to genomic endpoints [109]. Web-based weight-loss trials revealed greater adherence when supported by personalized digital coaching, a domain primed for AI automation.

### 8.6. AI in Reproductive and Immune Medicine: Precision Fertility, Immunotherapy, and Rare Diseases

Endometrial immune profiling combined with tailored therapy increased live birth rates after embryo transfer, an outcome AI could optimize via stratification algorithms [110].

Denosumab stimulated spermatogenesis in infertile men with intact Sertoli cell capacity, presenting a case for AI to identify suitable endocrine phenotypes [111].

IgE-guided elimination diets in eosinophilic esophagitis showed precision beyond pharmacology, and AI could refine dietary personalization by immune profile [112].

Semi-individualized Chinese medicine regimens demonstrated pragmatic personalization in diabetic nephropathy, providing culturally diverse training data for AI [113].

A multicenter Chinese RCT on recurrent urinary tract infections validated individualized antibiotic and herbal treatments, underscoring AI’s potential in non-Western modalities [113].

### 8.7. Implementation Challenges: Data Integration, Bias, and Equity

Community health workers delivering precision interventions show that personalization must be embedded in context-specific health systems, not only high-tech hospitals [114]. Measuring treatment response robustly in multiple sclerosis provided multidomain endpoints, critical for validating AI predictions [115]. AI-enabled adaptive trial structures, such as umbrella and N-of-1 designs, produce high-resolution training data while matching patients to optimal sequences [103].

### 8.8. Future Directions: Digital Twins, Adaptive Trials, and Global AI-Powered Personalization

Multiscale PK/PD modeling in breast cancer anticipates the rise of “digital twin” patients whose therapy can be simulated before administration [116]. Personalized tacrolimus dosing in liver transplantation shows that phenotype-driven AI can outperform standard care in pharmacology [116]. Nortriptyline dosing optimized via pharmacogenetics and phenotyping illustrates how AI harmonizes molecular and clinical predictors to refine psychopharmacology [91]. Ultimately, AI must evolve from retrospective modeling to prospective, globally validated tools, integrating diverse datasets while reducing bias and ensuring equitable access [91].

## 9. Personalized Therapeutic Strategies for Athletes

Precision sports medicine is rapidly transitioning from theory to practice, driven by the convergence of multi-omics, high-resolution physiological monitoring, and data-guided clinical decision-making. This section provides an integrated biological rationale and a translational roadmap for implementing precision interventions in competitive athletes, emphasizing molecular mechanisms, phenotyping frameworks, and governance infrastructures essential for safe, scalable adoption.

### 9.1. Biological Foundations of Individualized Athletic Adaptation

Exercise generates an orchestrated cascade across metabolic, neuromuscular, endocrine, and immune systems, with each domain contributing to how an athlete adapts or maladapts to training stimulus. Canonical molecular pathways, most prominently mTOR, AMPK, PGC-1α, and downstream mitochondrial regulators, govern substrate selection, mitochondrial biogenesis, and muscle remodeling, creating individualized metabolic signatures after training [117].

Multi-omic profiling demonstrates that exercise induces reproducible tissue-specific modifications across transcriptomic, proteomic, lipidomic, and epigenomic layers, a framework described comprehensively in the MoTrPAC atlas, which serves as a reference for distinguishing physiologic adaptation from early maladaptation [118].

Immune engagement, including NF-κB activation, cytokine cascades, and leukocyte trafficking, is essential for inflammatory resolution and subsequent muscle repair, and the magnitude and timing of these responses vary substantially between individuals [119].

Inter-individual variability in these networks is shaped by genetic polymorphisms, epigenetic marks, biological sex, age, training history, metabolic phenotype, and sleep/stress status. These determinants explain why athletes display heterogeneous responsiveness across strength, endurance, neuromuscular adaptation, and recovery kinetics [120]. Therefore, personalized strategies must begin with a mechanistic understanding of how molecular signals integrate with physiological outputs to pre-empt overtraining, optimize recovery, and guide intervention timing.

### 9.2. Integrated Precision Interventions: From Phenotyping to Management

#### 9.2.1. Mechanism-Informed Phenotyping

A robust baseline phenotyping protocol incorporates clinical evaluation, physiological testing, neuromuscular performance metrics, psychological screening, and selective multi-omic profiling. High-quality, comprehensive baselines are essential because adaptive or maladaptive signals are only interpretable relative to an athlete’s individualized reference point [120].

Cardiopulmonary assessments, including resting and maximal VO_2_, ventilatory thresholds, and lactate kinetics, identify aerobic efficiency, recovery capacity, and substrate utilization patterns. Neuromuscular characterization requires isometric and isokinetic strength mapping, rate-of-force-development analysis, and fatigue indices to quantify mechanical resilience and asymmetries [121].

Psychological and sleep assessments provide essential context for interpreting molecular or physiological fluctuations, as mood disorders, poor sleep, and stress markedly affect inflammatory reactivity, muscle recovery, and hormone regulation [122].

Selective genomics, proteomics, and metabolomics are justified when the results guide actionable decisions, for example, ferritin-based iron optimization, inflammatory risk profiling, or genotype-informed caffeine dosing. Integrating longitudinal wearable data with these phenotypes enhances resolution; deviations from individual baselines, not population norms, drive clinical interpretation [123].

#### 9.2.2. High-Fidelity Digital Monitoring and Biomarker Integration

Continuous physiological monitoring has become central to athlete management. However, device validity varies dramatically. Research-grade ECG chest straps, validated inertial measurement units, and high-frequency GPS outperform most wrist-based PPG systems, particularly when clinical decisions involve cardiac or neuromuscular load [124]. ECG-derived HRV remains the gold standard, whereas wrist-derived pulse-rate variability frequently diverges under motion, reduced perfusion, or artifact [125].

Wearable interpretation must incorporate contextual modifiers, including travel fatigue, illness, sleep debt, environmental stressors, and competition density [126].

Fusing digital data with periodic biochemical panels, including CK, CRP, IL-6, cortisol, ferritin, and targeted metabolomics, improves sensitivity for detecting overreaching, immune suppression, or insufficient recovery. Algorithms that flag statistically significant deviations from individualized baselines allow early, mechanism-guided interventions such as load modification, nutritional correction, or targeted rehabilitation [127].

#### 9.2.3. Precision Nutrition and Metabolic Optimization

Precision nutrition corrects measurable deficiencies, optimizes macronutrient timing, and aligns fueling strategies with metabolic phenotype. Biochemical assays (vitamin D, ferritin, electrolytes) guide targeted correction, as deficiencies in iron and vitamin D strongly impair performance and recovery [128].

Metabolic phenotyping via indirect calorimetry and substrate oxidation testing informs individualized carbohydrate and fat utilization strategies during training and competition. Indiscriminate supplementation is discouraged due to lack of efficacy or potential toxicity. Where evidence is robust, nutrigenomics provides meaningful stratification: CYP1A2 variants shape caffeine metabolism, influencing ergogenic benefit and risk. Individuals with the AA genotype, who are fast caffeine metabolizers, tend to show consistent performance improvements with caffeine supplementation. In contrast, carriers of the C allele may derive smaller benefits or even reduced effects, highlighting the value of genotype-based personalization in sports nutrition [129].

#### 9.2.4. Pharmacogenomics and Anti-Doping Integration

Pharmacogenomic profiling involving CYP2C9, CYP2D6, transporter polymorphisms optimizes medication safety for analgesics, NSAIDs, opioids, and psychotropics [130].

Genotype-guided prescribing reduces adverse effects and improves recovery trajectories when aligned with evidence-based guidelines [131].

All medications must be cross-checked against the WADA Prohibited List. When essential drugs are classified as prohibited, clinicians must submit comprehensive Therapeutic Use Exemption (TUE) documentation. Embedding TUE workflows into routine clinical practice protects athlete safety and regulatory compliance (USADA, 2001).

#### 9.2.5. Individualized Injury Prevention and Rehabilitation

Biomechanical screening, including movement quality, asymmetry mapping, strength ratios, hop tests, and load-management analytics, forms the backbone of injury prevention.

Where validated, genetic susceptibility markers (such as collagen gene variants) help stratify tendon and ligament injury risk [132].

Rehabilitation should follow criterion-based rather than time-based progression, incorporating functional testing, biomarker normalization, wearable-verified movement quality, and neuromuscular control indices.

This approach consistently reduces re-injury rates and enhances return-to-play readiness [133].

#### 9.2.6. Psychological Health and Sleep as Adaptive Modifiers

Sleep quality and mental health modulate immune function, hormone regulation, and recovery speed [134].

Psychological tools and sleep/HRV monitoring detect early signs of stress accumulation, burnout, and maladaptation. Personalized interventions, including sleep hygiene protocols, stress-periodization, and cognitive-behavioral approaches, are essential therapeutic elements [135].

### 9.3. Implementation, Governance, and Translational Roadmap

Operationalizing precision athlete care requires multidisciplinary teams, standardized monitoring protocols, and athlete-centric informed consent with explicit genomic and digital data considerations. Data governance frameworks must ensure encrypted storage, restricted access, athlete data ownership, and transparency regarding secondary data use to prevent misuse or discrimination [136].

Major evidence gaps persist: lack of large prospective multi-omic/wearable cohorts, few personalized RCTs, inconsistent wearable validity, and limited reproducibility of gene–trait associations [137].

Pilot programs with iterative validation in sport-specific environments are recommended prior to large-scale deployment [138].

Practical clinical recommendations include constructing individualized baselines, using research-grade devices for high-stakes decisions, embedding WADA/TUE checks into prescribing, and adopting criterion-based rehabilitation frameworks [139].

### 9.4. Sports Genotyping and DTC Testing

The application of genetic testing in sports performance remains controversial. Recent expert reviews and position papers emphasize that current evidence does not support the use of genetic or polygenic markers for talent identification or performance prediction, citing limited predictive validity and poor translational value into training decisions [140]. Direct-to-consumer (DTC) genetic testing has been specifically criticized for oversimplification of complex traits, lack of clinical oversight, and potential ethical risks, including misinterpretation, psychological harm, and discriminatory practices [141]. Accordingly, genetic data in sports should be confined to clinically relevant contexts and interpreted by qualified professionals within robust ethical and regulatory frameworks.

## 10. Challenges in Implementing Personalized Medicine

Implementing personalized medicine faces multiple interrelated challenges spanning technical, economic, workforce, ethical, and practical domains. Technical and data integration barriers are prominent, such as managing and integrating vast amounts of multi-omics data, including genomics, transcriptomics, proteomics, metabolomics, and epigenomics, which requires advanced computational techniques, extensive storage capacity, sophisticated analytical strategies, and virtual processing capabilities [142]. These challenges contribute to the slowness of healthcare research and hinder the creation of tailored care due to difficulties in connecting and efficiently utilizing health information across multiple sites.

This is often attributed to persistent interoperability and data integration barriers, fragmented systems, inconsistent standards, and limited cross-platform data exchange, which hinder comprehensive data use in multisite research and personalized care delivery [141].

Economic and access issues further complicate implementation, since personalized medicine relies on high-cost technologies such as high-throughput sequencing platforms and specialized bioinformatics tools, which demand substantial funding often unavailable in many countries and healthcare systems [142]. In addition, workforce and training gaps persist, as successful adoption depends on well-educated patients, laboratory technicians, IT specialists, and healthcare professionals. Many organizations lack the infrastructure and support to transition from standard practices to patient-centric approaches [141].

Regulatory and ethical concerns also pose obstacles, including misconceptions about personalized medicine, unrealistic expectations, and the need to raise public awareness regarding its true capabilities [142]. Finally, the implementation in clinical practice varies significantly between rural and urban areas, as each setting has distinct requirements and faces unique logistical and infrastructural challenges [134]. Collectively, these barriers highlight the complexity of translating personalized medicine from research to routine clinical care.

Beyond technological constraints, implementation barriers substantially limit the real-world impact of personalized medicine. High upfront costs extend beyond sequencing to bioinformatics, follow-up, and clinician training, challenging reimbursement even in high-income healthcare systems [140]. Fragmented regulatory frameworks delay approval and standardization, while in low- and middle-income countries, limited infrastructure and workforce capacity restrict access, reinforcing global disparities [132]. Addressing these challenges requires coordinated regulatory reform, sustainable financing, and workforce education, rather than reliance on technology alone [133].

### Health Economics and Cost-Effectiveness of Personalized Medicine

Health economics helps us understand not only whether personalized medicine (PM) works, but whether it delivers real value for patients and healthcare systems. Through health technology assessment (HTA), PM interventions are examined from multiple angles: how well they improve health, what they cost, and the ethical or organizational challenges they raise. Economic evaluations, such as cost-effectiveness analyses using incremental cost-effectiveness ratios (ICERs) or incremental net monetary benefits (INMBs), allow us to compare personalized treatments with standard care in a structured way. While many PM approaches do improve patients’ quality of life, these gains do not always translate into good value for money. In many cases, large numbers of people must undergo expensive testing to identify the small group that can benefit from a treatment, thereby raising overall costs. This is especially true for interventions targeting rare genetic markers, such as NTRK fusions, where widespread screening contributes to high ICERs despite strong benefits for the few eligible patients. Added to this, value-based pricing can push treatment costs even higher. As a result, even when PM offers meaningful clinical benefits to individuals, its economic value ultimately depends on how common the targeted biomarker is, how costly the testing process becomes, and whether the health gains outweigh the financial burden placed on the healthcare system [140].

## 11. Limitations, Failures, and Translational Gaps in Personalized Medicine

Despite substantial progress in precision medicine, multiple limitations constrain its routine clinical implementation. While whole-genome sequencing (WGS) offers comprehensive detection of genomic variation, its diagnostic yield varies widely across clinical contexts and is often lower in polygenic, multifactorial, and poorly phenotyped conditions such as complex neuropsychiatric disorders. Meta-analyses indicate that while WGS may modestly increase overall diagnostic yield compared with targeted methods in rare diseases, the incremental yield beyond more established tests remains variable and modest, reflecting challenges in variant interpretation and clinical utility in broader populations.

WGS implementation also faces technical, interpretative, and infrastructure barriers: the large data volumes demand robust computation and specialized interpretation expertise, and there are no universally accepted standards for non-coding or low-penetrance variant interpretation, reducing actionable outcomes in many cases [141]. In addition, existing genomic reference datasets are heavily biased toward individuals of European ancestry, undermining generalizability and accuracy for underrepresented populations and exacerbating health disparities in genomic diagnostics [120].

High-profile translational challenges in AI also underscore the practical limits of prematurely deployed clinical decision tools. For example, IBM Watson for Oncology (WFO), launched with expectations as an AI-driven decision support platform, demonstrated variable concordance with expert oncologist recommendations across cancer types (often well below perfect agreement) and raised concerns about real-world contextual awareness and data representativeness. Studies of WFO reported concordance rates ranging approximately 64–96% depending on cancer type and setting, highlighting inconsistencies across clinical contexts and the need for extensive validation before clinical uptake [141].

Reviews of WFO note that while promising as a decision-support adjunct, significant limitations remain in interpretability, localization to practice patterns, and consistent utility, limiting its ability to replace clinician judgment [142].

Beyond technology performance, system-level barriers hinder adoption: high costs of advanced genomic testing, insufficient reimbursement frameworks, limited access to specialized testing in low-resource settings, and regulatory gaps for AI and genomic applications create disparities in access and integration into standard care [141]. Collectively, these factors emphasize that technological promise alone is insufficient; rigorous clinical validation, transparent workflows, data diversity, affordability, and system readiness are essential prerequisites for meaningful clinical impact.

## 12. Ethical, Legal, and Social Implications of Personalized Medicine

The expansion of personalized medicine raises critical ethical, legal, and social considerations that must be explicitly addressed to ensure responsible implementation and societal trust. Informed consent in genomic testing, especially for whole-genome sequencing, is uniquely complex because secondary and incidental findings may emerge that are unrelated to the primary diagnostic intent. These findings can have significant psychological and clinical implications unless patients receive thorough pre-test counseling and retain autonomy over which results they wish to receive, reinforcing the need for robust consent processes tailored to genomic contexts [140].

Data privacy and governance represent foundational challenges in personalized medicine. Genomic and digital health data are inherently identifiable and potentially long-lasting, leading to heightened risks of misuse or re-identification if protections are inadequate. This demands compliance with stringent regulatory frameworks such as the General Data Protection Regulation (GDPR) (in the EU) and parallel data-protection standards to protect confidentiality and prevent unauthorized access or discrimination [142].

Equity remains a central ethical concern. High costs, limited infrastructure, and workforce shortages risk concentrating on the benefits of precision medicine within high-income settings, thereby widening global and regional health disparities. Without deliberate policy interventions to promote inclusion, underrepresented populations may face reduced access, further entrenching inequities in healthcare outcomes [141].

Addressing these ethical, legal, and social implications requires transparent governance structures, equitable access strategies, and ongoing public engagement, as well as institutional frameworks that balance innovation with respect for individual rights and societal values [140].

## 13. Conclusions and Policy Recommendations

Personalized medicine has moved past its expected promise and is now changing clinical practice. Genomics, multi-omics, wearable physiology, and machine learning are opening the door to more timely diagnoses, better drug prescribing, adaptive oncology, and individualized training and recovery to support patients through pregnancy, pediatrics, adult care, geriatrics, and sportomics. These technologies provide reproducible clinical and performance benefits, but they remain limited mainly to elite centers and pilot programs. To translate these benefits into broad evidence and equitable impact, we need to be coordinated in advancing on four interrelated and synergistic priorities: representative validation to eliminate bias and establish generalizability, interoperable and privacy preserving architectures for secure federated sharing, clinician and coach-centered implementation (workflows, education, decision support) to align with real world practice, and scalable and cost-sensible platforms to enable adoption in low-resource environments. When combined with transparent governance, prospective clinical evaluation, and incentives for equity, these components can facilitate the formalized transition of personalized medicine from elite practice to the standard of care and performance optimization. The true test of success in the coming decade will not be the gadgets we invent or advanced algorithms that do amazing things, but rather the degree to which we will make this new science work reliably, equitably, and broadly to improve lives. But bridging these scientific revolutions to the national implementation will require a clear policy vision implemented with active collaboration across public and private sectors. Our vision of public impact will ultimately require engagement from health, consumer technology, education, and regulatory domains.

### Recommendations for Policy and Decision-Makers

Coordinated action across key governmental sectors is essential to scale personalized medicine and AI-enabled healthcare. The Ministry of Health should set standards for integrating AI diagnostics, biosensor data, and multi-omics analytics, ensuring accuracy, validation, and equitable access. The Ministry of Communications and IT must provide secure, interoperable digital infrastructures to support large-scale data integration, federated learning, and evidence-based medicine. The Ministry of Higher Education and Scientific Research should promote interdisciplinary education and translational research to bring laboratory discoveries into clinical practice. Regulatory and funding agencies must establish ethical, legal, and financial frameworks, support innovation while protecting vulnerable populations. Together, a unified digital health strategy can expand personalized medicine nationally, enhance health outcomes, and reduce disparities.

## Figures and Tables

**Figure 1 diseases-14-00012-f001:**
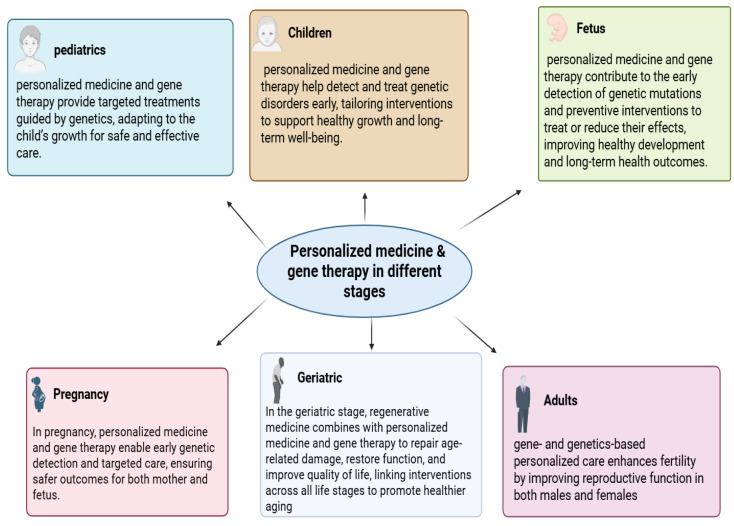
The role of personalized medicine and gene therapy across different life stages. Personalized strategies provide early detection, targeted treatments, and preventive interventions during the fetal and pregnancy stages, while continuing to support health, development, and quality of life in pediatrics, childhood, adulthood, and geriatrics.

**Figure 2 diseases-14-00012-f002:**
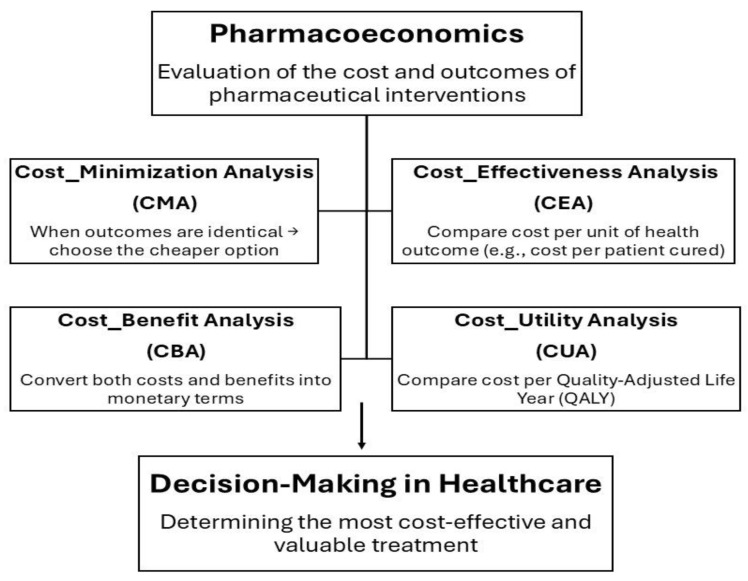
Overview of the Pharmacoeconomic Evaluation Framework showing the four main analysis types used to compare costs and outcomes of pharmaceutical interventions.

**Table 1 diseases-14-00012-t001:** Pharmacogenomic Biomarkers and Their Clinical Applications.

Biomarker	Drug	Therapeutic	Application	Reference
HLA-B 5701	Abacavir	HIV/Antiretroviral therapy	Screening HLA-B5701 before Abacavir reduces the risk of severe hypersensitivity reactions.	[40]
CYP2C19 (poor/intermediate metabolizer variants)	Clopidogrel	Cardiology/antiplatelet therapy	Identifying poor metabolizers allows alternative drug choice or dose adjustment to ensure efficacy and safety.	[41]
TPMT (and sometimes NUDT15)	Azathioprine, 6-Mercaptopurine	Oncology/immunosuppression/autoimmune disease	Genotyping TPMT (and sometimes NUDT15) before thiopurines helps identify patients at risk of severe myelosuppression → dose reduction or alternative therapy.	[42]
DPYD (deficiency or reduced-function variants)	5-Fluorouracil, Capecitabine	Oncology/chemotherapy	DPYD genotyping before fluoropyrimidine therapy can predict the risk of severe toxicity, allowing dose adjustment or alternative therapy.	[43]
UGT1A1 (reduced-function alleles, e.g., 28 variants	Irinotecan	Oncology/chemotherapy	UGT1A1 genotyping predicts the risk of severe irinotecan-related toxicity (neutropenia, diarrhea) and guides dose adjustment or alternative therapy.	[44]

## Data Availability

The original contributions presented in this study are included in the article. Further inquiries can be directed to the corresponding authors.

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
